# Ambiguous Results When Using the Ambiguous-Cue Paradigm to Assess Learning and Cognitive Bias in Gorillas and a Black Bear

**DOI:** 10.3390/bs7030051

**Published:** 2017-08-09

**Authors:** Molly C. McGuire, Jennifer Vonk, Zoe Johnson-Ulrich

**Affiliations:** Department of Psychology, Oakland University, Rochester, MI 48309, USA; mcmcguir@oakland.edu (M.C.M.); zjohnson@oakland.edu (Z.J.-U.)

**Keywords:** gorilla, black bear, cognitive bias, ambiguous-cue, learning

## Abstract

Cognitive bias tests are frequently used to assess affective state in nonhumans. We adapted the ambiguous-cue paradigm to assess affective states and to compare learning of reward associations in two distantly related species, an American black bear and three Western lowland gorillas. Subjects were presented with three training stimuli: one that was always rewarded (P), one that was never rewarded (N) and one that was ambiguous (A) because its reward association depended on whether it had been paired with P (PA pairing) or N (NA pairing). Differential learning of NA and PA pairs provided insight into affective state as the bear and one gorilla learned NA pairs more readily, indicating that they focused on cues of reinforcement more than cues of non-reinforcement, whereas the opposite was true of one gorilla. A third gorilla did not learn either pairings at above chance levels. Although all subjects experienced difficulty learning the pairings, we were able to assess responses to A during probe trials in the bear and one gorilla. Both responded optimistically, but it was difficult to determine whether their responses were a true reflection of affective state or were due to preferences for specific stimuli.

## 1. Introduction

The ambiguous-cue paradigm (ACP) was originally developed to assess mechanisms underlying learning [1,2], but can be adapted to assess emotional states in nonhumans (e.g., [3]). The task is simple in its design, yet presents subjects with a discrimination that many find difficult to learn (e.g., [4]). The paradigm presents the subject with a simultaneous discrimination task involving two pairs of stimuli. One stimulus (positive or P) is always reinforced when selected, one stimulus (negative or N) is never reinforced, and one stimulus (ambiguous or A) is reinforced depending on whether it has been paired with the positive (PA pairings) or negative stimulus (NA pairings). Although many species have demonstrated better learning of the NA compared to the PA pairing (e.g., children and the mentally disabled [5]; rhesus monkeys (*Macaca mulatta*) [6,7,8]; pigeons (*Columba livia domestica*) [9,10]; European starlings (*Sturnus vulgaris*) [11]), rhesus monkeys have also displayed superior learning on PA trials [6,7,12], as have chimpanzees (*Pan troglodytes*, [2]), and pigeons [9]. These different patterns of learning may be informative both in terms of learning strategy and cognitive biases, which may relate to affective disposition.

NA trials may be easier to learn compared to PA trials because the PA trials present an “approach-approach conflict” [11] as P is always reinforced and A is reinforced half of the time, leaving the subject conflicted over which reinforced stimulus to choose. However, one could argue that NA pairings involve an avoid/avoid conflict as they present subjects with one stimulus that is never reinforced and one that is reinforced only half the time. Thus, animals’ responses to the NA and PA pairings may indicate whether they attend more to cues of reward or to non-reward, and thus, might be useful in assessing their affective states. Animals that focus on cues of reward may be influenced more strongly by the approach-approach conflict and so learn NA faster (where only one stimulus has been rewarded), whereas animals that focus on cues of non-reinforcement may experience an “avoid/avoid” conflict and find PA easier to learn. Indications of focus on reward or non-reward cues may align with positive or negative affective states. 

In humans, affective states can be indicated by cognitive biases that are demonstrated by the preferential processing of certain types of information, such as threatening stimuli in the case of a negative bias [13]. The judgments of animals, just like their human counterparts, may be influenced by their emotional states [14,15]. An animal that is shown to react to ambiguous stimuli when presented under novel conditions similar to the manner in which they previously responded to rewarded stimuli may be seen as behaving optimistically. Alternatively, an animal that is shown to react to ambiguous stimuli similar to the manner in which they previously responded to non-rewarded or aversive stimuli can be seen as behaving pessimistically [15]. Pessimistic or optimistic affective states can be evoked by manipulating environmental conditions. 

There is evidence that providing animals with larger, more enriched enclosures may elicit positive cognitive biases (in rats [16]). Captive European starlings exhibited more optimistic interpretations of ambiguous stimuli after they had been housed in larger cages that also contained more enrichment items (such as water baths, perches, and bark chips) compared to when they had been housed in smaller cages lacking any additional enrichment items—although, in this case, space and enrichment were confounded [17]. The cyclical changes in environment that captive animals encounter may be due to the seasonal changes in climate, seasonal changes in visitor numbers, or seasonal changes in husbandry routines. In Experiment 1 of the present study, three male gorillas experienced changes in the amount of space they were offered, and arguably the types of enrichment available, over the course of a year. It may be that these seasonal changes in the gorilla habitat cause similar patterns of cognitive bias as those described by Matheson et al. [17], with the gorillas displaying more positive biases when in the outdoor (larger) habitat than when in the indoor (smaller) habitat. 

In Experiment 2 of the present study, the American black bear did not experience changes to habitat size, but rather changes to visitor density, with warmer weather during the summer drawing larger crowds than cooler temperatures during the fall and winter. To date, there has been only one study to investigate cognitive biases in bears. Keen et al. [18] tested grizzly bears (*Ursus arctos horribilis*) on a novel cognitive bias task that made use of differential distribution of food rewards. The bears in this study were trained to respond differently (touch with a nose or a paw) to two different stimuli (a light grey cue card or a dark grey cue card) in return for either a large or small amount of food. Following training, the bears were exposed to enrichment items that varied in preference. After the enrichment exposure, the effects were assessed by presenting probe stimuli (intermediate shades of grey) to the bears and observing whether they responded in a manner that corresponded to larger amounts of food (optimism) or a smaller amount of food (pessimism) during training. Keen et al. [18] did not detect any effect of enrichment type on the bears’ cognitive biases during testing. However, they did observe that when the bears spent more time engaged in anticipatory behaviors (i.e., pacing) prior to testing, they displayed positive cognitive biases (optimism). It may be that a two-hour acute exposure to enrichment items (a cow hide and a parking cone) was not sufficient to induce a lasting change in cognitive bias. It is possible that other environmental factors, such as seasonal changes, including visitor density, may have a more lasting impact on cognitive biases. 

We previously used a modified version of the ambiguous cue paradigm to assess affective state in western lowland gorillas (*Gorilla gorilla gorilla*) but found the training period necessary to coincide with manipulations of browse foraging enrichment [19] too brief to allow for adequate learning of the discriminations [3]. In the current study, we extended training of the ambiguous cue paradigm in the gorillas (*Gorilla gorilla gorilla*) and applied the same paradigm to cognitive bias assessments in an American black bear (*Ursus americanus*). In both cases, we used the ambiguous cue paradigm as a means to assess affective state in these subjects as part of a long-term welfare assessment. As an alternative to paradigms that present probe stimuli that are intermediate in some stimulus dimension along a continuum between reward and nonreward stimulus properties (e.g., [20]), we used the ambiguous stimulus presented in the simultaneous discrimination task paired with a novel stimulus as a means to assess optimism and pessimism. If the ambiguous cue is chosen over the novel cue, this indicates an optimistic attitude toward a stimulus to which responses have been reinforced and non-reinforced equally often. This paradigm has the advantage that it does not involve the presentation of intermediate stimuli such that responses may simply reflect a perceptual discrimination of stimuli closer to reward and non-reward contingencies. Furthermore, performance on the training pairs indicate whether animals attend to avoid/avoid or approach/approach conflicts, and thus, can also shed light on potentially stable cognitive biases/affective dispositions in individuals.

The experiments described below present a rare opportunity to compare acquisition and mastery of the ambiguous-cue paradigm in a bear and in gorillas, given previous studies suggesting that bears perform as well as, if not better than, great apes on cognitive tasks, such as the discrimination of natural categories [21,22,23,24,25], and quantity estimation [21,26]. The capacity of bears to outperform apes in cognitive tests supports recent conjecture that foraging complexity is potentially more important than sociality in driving the evolution of certain aspects of complex cognition [27,28] given that bears experience low levels of sociality but varying levels of foraging complexity whereas apes experience high levels of sociality and foraging complexity. Both species can be described as generalists that exploit a patchily distributed diet and engage in extractive foraging [29], qualifying them as experiencing complex foraging demands. We were interested in cognitive bias at the individual level and all subjects were of interest in this regard given their unique housing situations (a bachelor group of gorillas and a solitary black bear).

## 2. Experiment 1

### 2.1. Materials and Methods

#### 2.1.1. Subjects

Three male silverback gorillas, Chipua (‘Chip’, 19 years old), Pendeke (‘Pende’, 18 years old) and Kongo (17 years old), were tested. These three half-brothers composed a bachelor group at the Detroit Zoo in Royal Oak, MI. Training and testing took place in an indoor housing area that is inaccessible to zoo visitors. The gorillas participated in training and testing sessions three mornings each week after they were separated for their morning feed. The gorillas participated in an unrelated conditional discrimination task during the same time period. Training and testing with both the American black bear and the Western lowland gorillas complied with the IACUC of Oakland University and provided a form of enrichment.

Testing was intended to coincide with seasonal changes in the gorillas’ habitat. In colder weather, the gorillas are restricted to their indoor habitat, which is much smaller than their outdoor habitat and has less natural enrichment items available. In warmer weather, the gorillas are often restricted to the outdoor habitat or given access to both the indoor and outdoor habitats. The outdoor habitat is larger and contains natural foraging opportunities in the form of plants (living grasses, bushes, and trees), in addition to many of the typical enrichment items that they may be offered in the indoor habitat (e.g., cardboard, toys, cut foraging material).

#### 2.1.2. Materials

The touchscreen apparatus consisted of a Panasonic Toughbook CF19 laptop computer and 19” Armorall capacitive touch-screen monitor (VarTech, Baton Rouge, LA, USA) welded inside a rolling LCD panel cart encased with top and sides. This apparatus was positioned so that it was within a fingers’ reach of the gorillas through the mesh of the enclosure. The experiment was programmed using Inquisit Version 3 (Millisecond, Seattle, WA, USA) for Windows. Pairs of stimuli were presented, to the left and right of the center of the screen. All stimuli consisted of arbitrary colored symbols drawn in Microsoft Word to ensure that the responses of the gorillas were not influenced by any prior associations (See Figure 1). During training, the gorillas were presented with three stimuli; one that was always rewarded (the positive stimulus, P), one that was never rewarded (the negative stimulus, N) and one that was rewarded when it was paired with N and not rewarded when it was paired with P (the ambiguous stimulus, A). These stimuli were presented in pairs in which A was paired with either N or P (the NA pairing and the PA pairing). The assignment of each stimulus to each role was counterbalanced across the gorillas with two additional novel stimuli presented during testing. Rewards across training and testing consisted of small food items present in the morning breakfast tray. 

#### 2.1.3. Training

The training sessions consisted of 10 trials of either NA or PA pairs presented in a two-alternative forced choice task. Three mornings each week, the gorillas received approximately four sessions of each discrimination. Each gorilla received approximately 200 sessions of each pairing. If they reached a criterion of 80% correct for four consecutive sessions, they would proceed to testing. 

#### 2.1.4. Testing

The original testing sessions consisted of ten trials of four PA, four NA, and two probe trials (presented on the fourth and eighth trials of the 10-trial sessions). The probe trials consisted of the ambiguous cue paired with one of the two novel cues to create two distinct probe pairings. The gorillas were rewarded for selecting either stimulus during probe pairings. 

Chip was the only gorilla whose performance approached our criterion of 80% correct, so he was presented with five test sessions during February, a time in which he was restricted to the indoors. In the final three sessions, two additional trials were included (for a total of 12 trials per session: four PA, four NA, two ambiguous probes on trials four and eight, and two familiarity control trials on trials three and ten) to assess whether Chip’s preference for A on the probe trials in the first two sessions was merely due to a preference for familiarity (A being familiar and the novel shapes unfamiliar). On the familiarity control trials, the familiar N and P items were also presented alongside novel stimuli to determine whether Chip’s selection of A in the earlier testing sessions was due to a preference for familiar versus unfamiliar items. These additional trials consisted of one N-novel pair and one P-novel pair. 

Given the gorillas’ inability to reach criterion after extensive training, testing was not conducted across multiple seasons as planned. Thus, testing never occurred in the warmer summer seasons.

### 2.2. Results

#### 2.2.1. Training

Figure 2 shows the performance of all three gorillas across 50 four-session blocks. Binomial tests on the final testing block indicated that Kongo performed at chance (50%) for both trial types on this task despite the extended training period (final block: *M*_NA_ = 50%, *SD* = 0.50, *p*_NA_ = 1.0, *M*_PA_ = 50%, *SD* = 0.50, *p*_PA_ = 1.0). His strong side bias led to consistent chance level performance given the counterbalancing of the stimuli. Binomial tests on Pende’s final testing block indicated that he was able to learn the NA pairing (final block: *M* = 92.5%, *SD* = 0.267, *p* = 0.00), but not the PA pairing (final block: *M* = 57.5%, *SD* = 0.50, *p* = 0.43), as indicated in Figure 2. Of the three gorillas, only Chip displayed adequate learning for both NA and PA trials (final block: *M*_NA_ = 75%, *SD* = 0.44, *p*_NA_ = 0.002, *M*_PA_ = 90%, *SD* = 0.30, *p*_PA_ = 0.00) in order to advance to testing. 

#### 2.2.2. Testing

Given the total number of trials Chip had received, and the fact that a seasonal change in weather was fast approaching, the testing criterion was relaxed to 75% correct allowing Chip to continue on to the testing phase of the study (Chip performed at 100% correct on two of his final four NA sessions although these sessions were not consecutive). Binomial tests across the trained trials during testing indicated that Chip selected the correct stimulus on the NA trials at a chance level (*N* = 20, observed proportion correct = 55%, *p* = 0.824) but he selected the correct stimulus on PA trials at a rate above chance (*N* = 20, observed proportion correct = 80%, *p* = 0.012). 

Across all five of the test sessions, when presented with the A-novel probe pairing, Chip selected A. Binomial tests indicated that Chip selected A (the optimistic choice) at a rate above chance (*N* = 10, observed proportion = 1, *p* = 0.002). Although Chip chose the familiar item on the ambiguous probe sessions, he did not exclusively select the familiar item on the familiarity control trials. Chip chose the familiar item 66.6% of the time (see Figure 3) during the familiar-novel trials and, of those trials, he selected P and N equally. Binomial testing indicated that Chip did not select the novel stimulus when it was paired with a familiar (P or N) item at a rate differing from chance (50%; *N* = 6, observed proportion familiar = 0.67, *p* = 0.688).

### 2.3. Discussion

Given that only one gorilla met the criterion for learning the discrimination during training, we were able to use the ambiguous/novel stimuli pairing to assess cognitive bias only in this one gorilla. Chip selected the ambiguous stimuli on all testing trials. If this performance (where he selected A more often than the novel stimulus) was due to an inherent preference for the A stimulus, Chip should have performed better on the NA (where A is rewarded) compared to PA pairs (where P is rewarded) across training. Instead, Chip displayed better learning of the PA pairing, suggesting that there was not an inherent preference for the A stimulus. In addition, although it was also possible that this finding was driven by an overall preference for familiar stimuli, subsequent familiar-unfamiliar pairings that included P and N suggest that this may not have been the case (see Figure 3). Although it is true that Chip selected the familiar stimulus (A) across all probe trials (A-novel), his selection of the familiar stimulus (P or N) on the P-novel or N-novel trials presented in the final three test sessions did not exceed chance levels. This finding, in conjunction with his relatively lower performance on NA trials, suggests that his responses on the A-novel pairings were not due to an inherent preference for familiar stimuli. Instead, Chip’s preference for A on probe trials could indicate optimism.

The weather during February when Chip was tested was cold—requiring that the gorillas be housed indoors. At the Detroit Zoo, the indoor habitat for the gorillas is smaller than the outdoor habitat, which means that for a portion of the year, the social group is compressed. We speculated that this possible optimism could be due to reduced anxiety or stress that could come from increased ease of monitoring the whereabouts of other group members in the smaller space. It is possible that Chip was displaying optimism during this period of restricted space, although it was not possible to train and test him on a separate set of stimuli during the summer months to clarify whether it was the environmental conditions (the fact that Chip was restricted to the indoor environment) that influenced his choices. 

Overall, it is clear from Figure 2 that it was difficult for the gorillas to learn the training pairs (NA and PA) and this made it difficult to assess cognitive bias using ambiguous/novel probes. Instead, we assessed learning of NA and PA trials in order to consider affective state with an alternative method. Chip and Pende displayed distinct differences in learning, with Pende learning NA better than PA and Chip displaying the opposite pattern. If the subjects learn NA parings faster than PA pairings, it could indicate that the subject is attending more strongly to reinforcement stimuli (interpreted as an optimistic response) whereas the opposite pattern may indicate that the subject is attending more strongly to non-reinforcement stimuli (interpreted as a pessimistic response). Thus, it is possible that Chip displayed a negative (or pessimistic) bias during training and Pende displayed a positive (or optimistic) bias during training.

Lastly, Kongo, whose responses were guided by a side bias throughout the experiment, remained at chance across the training phase for both types of trials. Therefore, Kongo’s performance tells us very little about the cognitive mechanisms underlying his choices. Given the difficulty in training the gorillas to learn match-to-sample and conditional discrimination tasks using two-dimensional stimuli (unpublished data), it is possible that they do not find such stimuli engaging or meaningful and are not motivated to learn associations with rewards given that they are not food adjusted for these studies. 

Given the generally quicker acquisition of cognitive testing by Migwan, an American black bear in other studies [30], we proceeded with testing her in the same paradigm.

## 3. Experiment 2

### 3.1. Materials and Methods

#### 3.1.1. Subjects

An unaltered 14-year-old female American black bear, Migwan, housed at the Detroit Zoo in Royal Oak Michigan was tested. Migwan was rescued as a wild cub when she was found injured. After her rescue, Migwan spent much of her time in the presence of people, and, as a result, appeared to be very interested and responsive to humans, especially her keepers. Since her rescue in 2003, she has lived at the Detroit Zoo. Currently housed alone, Migwan was housed socially with one or two conspecifics until 2011. Migwan had previously been trained on various touchscreen computer tasks including training on a two-alternative forced-choice task, and a preference assessment task involving conditional discriminations similar to that taught to the gorillas (unpublished data), as well as an object recognition task [31].

#### 3.1.2. Testing Environment

Migwan’s enclosure consisted of an outdoor habitat (50’ × 100’) and an indoor off-habitat holding area. Research took place in this off habitat holding area. Food rewards consisted of a small portion of her diet that was set aside for training. Migwan was typically fed once in the morning before the zoo opened and once at the end of the day.

Just as the gorillas experience seasonal changes in their environment, Migwan experienced seasonal changes in visitor density, with warmer weather during the summer drawing larger crowds than cooler temperatures during the fall and winter. Migwan did not experience changes in habitat access, but it may be that certain environmental characteristics associated with smaller visitor numbers, such as reduced ambient noise, may also have a positive effect on animals’ affect. Furthermore, Migwan experienced a period of semi-hibernation with reduced enrichment and foraging opportunities, as well as activity levels, during the colder winter months.

#### 3.1.3. Materials

An ASPIRE One netbook with built-in touchscreen and a 19” VarTech Armorall capacitive touch-screen monitor, similar to the one described in Experiment One, was used. The experiments were programmed using Inquisit 3.0 for Windows. Two stimuli (2340 × 4160 MP) drawn in Microsoft Word were presented simultaneously and were centered to the left and right of the center of the screen. Because Migwan was presented with two phases of cognitive bias testing, she was trained and tested on a different set of stimuli for each phase (for a total of two sets of five stimuli, see Figure 1) to minimize the chances of her learning to associate the novel probe stimuli with a specific outcome. Ten arbitrary symbols (five in each set) were used to avoid any prior associations or biases with known shapes or objects. Correct responses (via nose touches) were followed by a melodic tone, a blank screen, and a food reward. Incorrect responses were followed with a blank screen, no auditory feedback, and no food reward.

#### 3.1.4. Training

A researcher set up the apparatus and software program while the keeper administered the food rewards to the bear. This experiment was timed to coincide with seasonal changes in zoo visitor density. We attempted to measure the degree of ‘optimism’ or ‘pessimism’ in the summer, when there was a high density of zoo visitors, and in the fall, when there was not. Migwan was trained approximately three days each week at midday. On each day of training, Migwan received three to six training sessions. She was brought in from the outdoor habitat for the duration of training and, after completing the training sessions, she was given access to the outdoor habitat again. 

To measure Migwan’s cognitive bias, the researchers trained her to discriminate between PA (positive and ambiguous) pairs and NA (negative and ambiguous) pairs. Migwan was presented with a two-alternative forced-choice task in which one cue was always the correct choice. Sessions consisted of eight trials: four PA and four NA trials presented in random order. For each phase of the study, Migwan was trained on a different cue set in case there were innate preferences for certain symbols that influenced her choices. Migwan was trained for the duration of the season and tested at the height of the season. For the summer phase, she received approximately 180 sessions and was tested in July when the weather was warm and visitor numbers are generally high. For the fall phase, she received 230 sessions. After 216 combined NA and PA sessions in the fall phase, separate NA and PA sessions were introduced (similar to the gorillas’ sessions described in Experiment 1). These separate sessions consisted of ten trials of only NA or PA pairings. Migwan completed an additional 14 sessions of each separate NA and PA pairing and was tested in October, shortly before she entered hibernation, when the weather was cooler and there were fewer visitors present. Migwan was not trained to a specific criterion; instead, similar to the methods used by Vasconcelos and Monteiro [11], Migwan was exposed to the same number of PA and NA trials across testing phases with the phases being timed to coincide with the end of the summer and fall seasons. 

#### 3.1.5. Testing

At the end of each season, Migwan participated in four testing sessions (i.e., two sessions per day over two consecutive days). Test sessions consisted of ten trials: two probe trials presented on the 4th and 8th trials, four PA and four NA trials, which were randomized in between. During the probe trials, Migwan encountered the original ambiguous stimulus paired with an additional novel stimulus. Testing followed the same procedure as described in Experiment 1 except that, there was no buzzer tone presented for incorrect choices, at the request of animal care staff. It was predicted that Migwan would display greater optimism during the fall months in which there was a decrease in visitor numbers. 

### 3.2. Results

#### 3.2.1. Training

As shown in Figure 4 and Figure 5, Migwan displayed the same pattern of learning in the summer phase and the fall phase, namely, that she learned the NA pairing (last 40 trials: summer *M* = 0.975, *SD* = 0.506; fall *M* = 0.900, *SD* = 0.304) better than the PA pairing (last 40 trials: summer *M* = 0.225, *SD* = 0.423; fall *M* = 0.500, *SD* = 0.506) . Binomial tests of Migwan’s last 10 sessions (or 40 trials) indicated that Migwan performed above chance (50%) on the NA pairing (*N* = 40, *M* = 0.98, *SD* = 0.158, *p* < 0.001), but that she performed significantly below chance on the PA pairing during the summer phase (*N* = 40, *M* = 0.23, *SD* = 0.423, *p* < 0.001). Binomial tests indicated that Migwan again performed above chance (50%) on the NA pairing (*N* = 40, *M* = 0.90, *SD* = 0.304, *p* < 0.001), but that she performed at chance on the PA pairing during the fall phase (*N* = 40, *M* = 0.50, *SD* = 0.506, *p* = 1.00).

#### 3.2.2. Testing

Binomial tests indicated that for the training pairs (PA and NA) presented during the test sessions, Migwan performed above chance (50%) on her NA pairing for both summer (*N* = 16, *M* = 100%, *SD* = 0.00, *p* < 0.001) and fall phases (*N* = 16, *M* = 100%, *SD* = 0.00, *p* < 0.001). Migwan performed significantly below chance on her PA pairing during the summer (*N* = 16, *M* = 18.6%, *SD* = 403, *p* = 0.021), whereas her choices did not differ from chance (*N* = 16, *p* = 0.454) for the fall PA pairing (*M* = 62.5%, *SD* = 0.5). 

Binomial tests indicate that Migwan did not select the ambiguous stimulus during testing trials at a rate different from chance (50%) for either phase (summer or fall). Figure 6, illustrates Migwan’s performance on the probe trials across the testing phases. During the summer phase, Migwan selected the ambiguous stimulus option 75% of the time. After completing the second phase of testing in the fall, it became apparent that Migwan had developed a bias for a specific stimulus, in this case one of the novel shapes (a gold star). When this preferred novel shape was displayed with A, she selected it. However, if A was displayed with the less preferred novel shape, she chose A. This made comparisons between the summer phase and fall phase problematic. Although it is difficult to draw conclusions about Migwan’s responses in the fall, during the summer phase, Migwan may have been optimistic, although not at a statistically significant level. Furthermore, her performance across training and testing in the summer suggest that she may have developed a preference for the ambiguous stimulus.

#### 3.2.3. Comparing Migwan’s Performance to That of the Gorillas

As Chip was the only gorilla to be tested on this task, his performance was compared to Migwan’s summer performance (her first round of training). Figure 7 shows the first 200 training trials of Migwan’s summer phase compared to Chip’s first 200 training trials (50 blocks of four sessions each). Figure 7 makes it clear that Chip and Migwan displayed opposite patterns of learning. Even though Figure 7 displays Migwan’s summer data, as can be seen in Figure 5, her fall data followed the same pattern. Although Migwan learned NA better than PA trials, Chip learned PA better than NA trials. However, Chip displayed improvement in his NA performance, allowing for testing. 

### 3.3. Discussion

We tested an American black bear on the ACP during two seasons to assess whether cognitive bias varied across seasons. Migwan performed equally well on both NA and PA pairings early on in training but, over time, as she learned the NA pairing, she seemed to struggle more with the PA pairing (Figure 4 and Figure 5). This pattern may indicate that the strategy she was using to perform correctly on the NA pairing was interfering with her ability to correctly respond during the PA pairing. Once she had learned the rule “chose A”, to solve the NA pairing, she seems to have used the same rule consistently, regardless of what stimulus A was paired with. The learning of the rule “chose A” may have also contributed to her preference for A during testing. 

Although this paradigm may prove to be useful in some cases, in this study, it was not possible to determine whether Migwan was behaving optimistically or pessimistically due to her innate preference for certain stimuli (in the first phase of testing, a green cross; in the second phase of testing, a gold star). It also became apparent that Migwan may have needed more time to learn the discrimination than was possible to provide during the course of this study. For these reasons, it is not possible to draw conclusions regarding her cognitive biases in her choices of A at testing. However, we did gain some insight into her learning patterns, specifically that she displays a pattern of learning (learning NA faster than PA) similar to other species tested previously [5,8,10,11], and the pattern displayed by Pende—one of the gorillas tested in Experiment 1. Chip, another gorilla subject, as well as rhesus macaques, chimpanzees and pigeons have displayed the opposite pattern of learning (learning PA faster than NA; [2,6,7,9,12]. However, as stated earlier, the patterns of learning displayed by these individuals may still provide insight into affective mental states. That Migwan more easily learned the NA discrimination suggests a positive (optimistic) bias across both seasons. 

Additionally, it is also possible that various daily changes may have influenced Migwan’s responding. Across this study, Migwan dealt with changes in visitor noise and density on a daily basis that could have influenced her performance on this task. As is true for any test administered in the zoo setting, Migwan was also exposed to the noise and sounds of the other animals in her building, which could have also influenced her mood or attention on any given test session. The small sample size of one bear also makes it difficult to generalize to other bears housed in similar circumstances. 

Finally, there was a limit to the number of trials Migwan could participate in during a single test session. Due to concern with her maintaining a healthy weight, there were a limited number of reward items that could be fed in a single day. Although extremely motivated by food, Migwan also appeared to be motivated by contact with humans (due to the fact that she was raised by people from a young age). This motivation was evidenced by her tendency to stay engaged with the keepers at the conclusion of the training periods (e.g., running back and forth along the dividing wall as if playing tag, splashing them with water when they left the area). The duration of the training sessions was dictated by the number of correct responses Migwan gave. The more incorrect choices she gave, the longer the session and vice versa. This resulted in a trade-off between access to food and access to staff. It may be that these conflicting motivations hindered Migwan’s learning as she might have associated non-reinforced responding with longer time spent with keepers.

Due to the fact that this method required extensive training (and a criterion level of performance was not met for both types of trials for any of our subjects), and the fact that innate preferences for shapes and colors influenced choices on probe trials, it was deemed necessary to train Migwan on a different cognitive bias assessment method, an active choice task, that was easier to learn and circumvented the problems of stimulus preference [30]. 

## 4. General Discussion

We used the ACP to test cognitive bias in gorillas and an American black bear. In general, all four subjects struggled to pass criterion for training, making testing with probe trials to assess cognitive bias difficult. However, we suggest that learning patterns during training may indicate long-term affective states. The individuals tested in these experiments displayed differences in their learning of the NA and PA pairs similar individual differences displayed in other species. Chip displayed superior learning for the PA pair. In contrast, Migwan and Pende’s performance on the NA pairing was better than their performance on the PA pairing. Pende’s performance on the PA pairing was poor enough that it actually prevented him from reaching criterion for testing. Migwan and Pende displayed a pattern of learning similar to that displayed by other species in previous studies [5,8,10,11]. Interestingly, like the gorillas in this experiment, rhesus macaques have shown different patterns of learning, both across studies and even within studies. The macaques in Fletcher and Garske’s [8] experiment learned the NA pairing faster, whereas the macaques in Boyer and Polidora’s [6] and Boyer et al.’s [7] experiments displayed the opposite pattern. Also interesting about these earlier experiments is that experimenters were able to reverse the pattern of learning displayed by the monkeys by changing aspects of the stimuli. In Boyer et al. [7], the macaques displayed the more typical pattern of learning NA faster than PA when using objects (three dimensional) stimuli instead of two-dimensional plaque stimuli. This aligns with a previous experiment involving the use of real objects [8]. Boyer and Polidora [6] were able to reverse the initial pattern of PA > NA by pretraining the monkeys using plaque stimuli with distinctive cues and then testing using stimuli with less distinctive cues. A similar result was obtained in pigeons [9]. When considering these studies together, it becomes clear that individuals show unique patterns of learning that are not species-specific and that the methods used in these experiments may greatly influence learning patterns within individuals. The distinctiveness of the A item may control whether learning is faster in PA compared to NA trials [9]. 

It is possible that differences in learning between the NA and PA pairings may shed some light on the affective state of these subjects. For instance, it is possible that Chip’s superior performance on the PA pairings compared to NA pairings could stem from the fact that he attended more to instances of non-reinforcement than reinforcement. If this were the case, the PA pairing would be easier to learn because A is at times non-reinforced whereas P is always reinforced (i.e., touches to P are never not rewarded). For an animal attending to non-reinforcement cues, PA is easier as there is only one cue present that is (at times) non-reinforced (A). During NA pairings, there are two cues present that, at times, are non-reinforced, making the discrimination more difficult. Typically, it is thought that individuals have an easier time learning the NA pairings as they are attending to instances of reinforcement and one stimulus in the NA pair is never rewarded (N), whereas the other is partially rewarded (A); the motivation to avoid touching N would align with the motivation to touch A. On the other hand, the PA pairing consists of two stimuli, both of which have been associated with rewards (P all of the time, A partially); the subject is faced with conflicting motivations to touch both stimuli [6]. Therefore, it is interesting that Chip learned the PA discrimination better than the NA discrimination. It may be that animals that are pessimistic attend more to non-reinforced stimuli whereas those that are optimistic attend more to reinforced stimuli. However, if Chip’s learning pattern does lend insight into his cognitive biases, the fact that he learned PA faster (attending to non-reinforced stimuli) indicating pessimism, would contradict his performance on the test phase of this study in which he selected A more often than the novel stimulus (indicating optimism). 

In contrast to Chip’s performance, Pende displayed superior learning for the NA trials compared to the PA trials. This pattern may indicate that he was attending more to the stimulus that was rewarded, and hence found the NA pairing easier to learn as N is never rewarded and A is at times rewarded. Again, for the NA pair, the motivation to avoid N aligns with the motivation to touch A, making this an easier discrimination compared to PA, in which the animal is faced with conflicting motivations to touch both stimuli. Migwan also displayed this same pattern of learning (NA > PA) suggesting that she, like Pende, attended more to cues of reinforcement than to cues of non-reinforcement. When tested with the ambiguous-novel probe pairings, Migwan displayed innate preferences for the stimuli—making interpretation of her test results problematic. 

Aside from the methodological issues, another challenge with assessing changes in cognitive bias across seasons is the difficulty in determining which factor might be responsible for any observed changes. That is, effects of visitor density, weather, and metabolic state tend to be confounded. Testing Migwan at the height of visitor season also required that she be tested in the middle of the summer, whereas testing during low visitor density required testing at the beginning of the fall season. American black bears undergo seasonal changes in metabolic rate and hormone levels associated with preparing for and enduring hibernation [32,33]. Including male or altered female bears in testing may be useful as they may experience less fluctuations in hormones associated with a naturally cycling female bear (although some changes are seen across the sexes as they prepare for hibernation). Testing could also be conducted under artificial seasonal patterns, where presentation of exhibit space or high visitor density could be balanced across seasons. In the future, it may also be useful to test for innate shape preferences before starting the training phase and to train to criterion on both trial types before proceeding to the testing phase. It would also be most beneficial to obtain other concurrent assessments of emotional state, such as hormone levels.

## 5. Conclusions 

It is clear that this task, especially the PA pairings, may be difficult for a wide range of species given that this pattern of learning has now been observed in Western lowland gorillas, American black bears, children, the mentally disabled [5], rhesus monkeys [8], pigeons [10], and European starlings [11]. Although the difficulty in learning both pairings makes this task difficult to modify for cognitive bias assessment, in which A is paired with novel stimuli, it still may be possible to assess cognitive bias by looking at learning patterns across the training phase of the ambiguous-cue paradigm. As such, we have demonstrated how a task designed to test learning mechanisms can potentially be extended to assess affective state in nonverbal animals.

## Figures and Tables

**Figure 1 behavsci-07-00051-f001:**
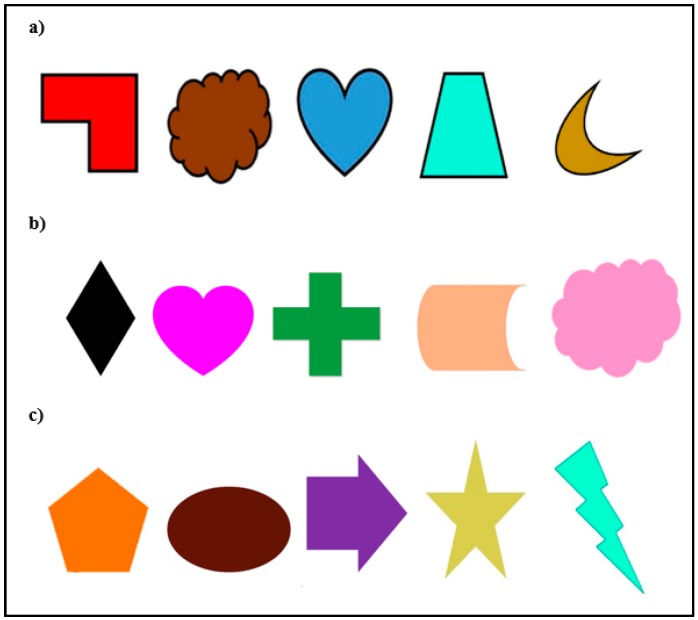
Stimuli sets (**a**) gorillas’ stimuli set; (**b**) Migwan’s summer set; (**c**) Migwan’s fall set. For each set, the three stimuli on the left are the training stimuli used to create the NA and PA sets (N, P, and A) while the two stimuli on the right are the novel stimuli introduced during probe trials. On each trial, the subject would see only two stimuli displayed next to each other on the screen according to which stimuli were designated as A, P and N for that subject.

**Figure 2 behavsci-07-00051-f002:**
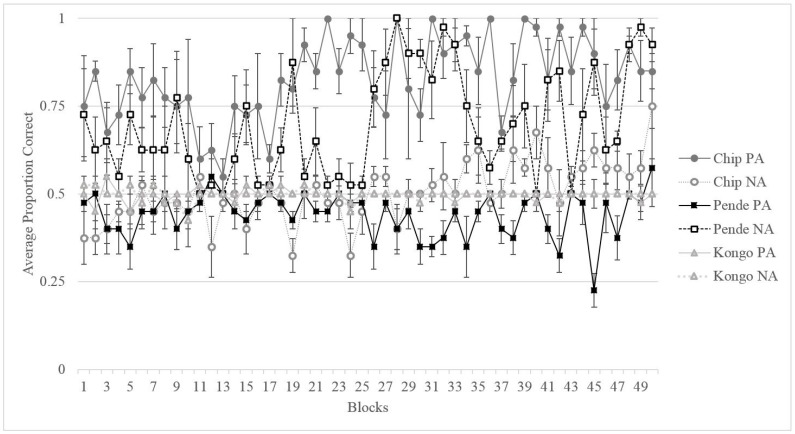
Average proportion of correct trials and standard error for both PA and NA trial types for each subject across the training phase in blocks of four sessions.

**Figure 3 behavsci-07-00051-f003:**
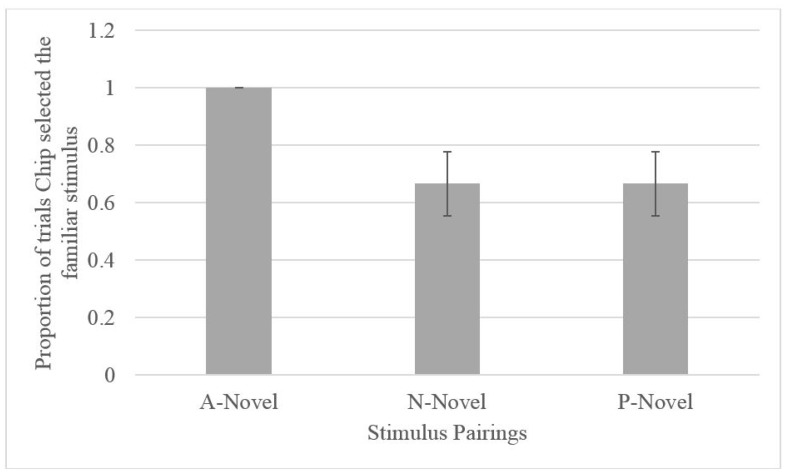
Proportion of testing trials on which Chip selected the familiar stimulus by stimulus pairings.

**Figure 4 behavsci-07-00051-f004:**
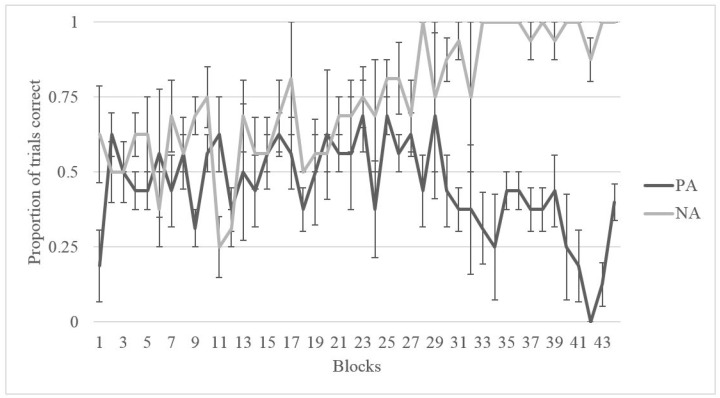
Proportion of correct trials for Migwan during training for both PA and NA pairs during the Summer Phase in blocks of four sessions.

**Figure 5 behavsci-07-00051-f005:**
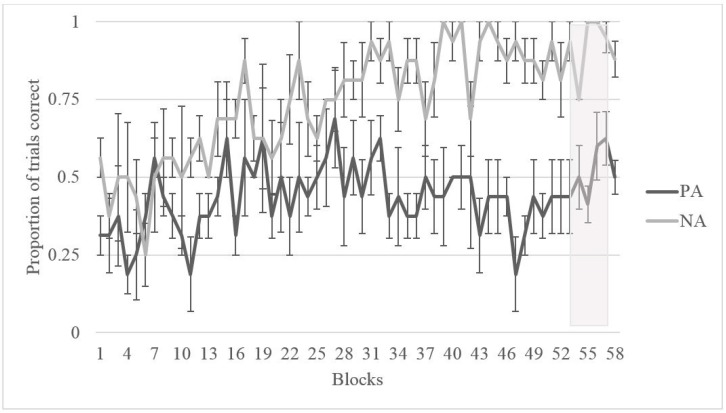
Proportion of correct trials for Migwan during training for both PA and NA pairs during the Fall Phase in blocks of four sessions—Shaded area indicating separate NA and PA sessions.

**Figure 6 behavsci-07-00051-f006:**
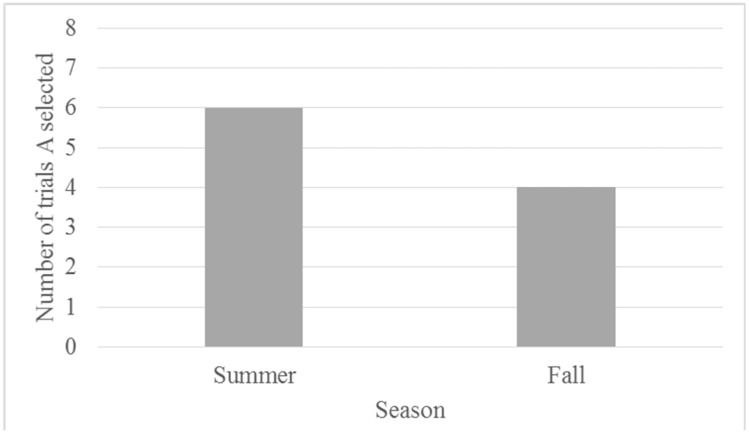
Number of trials (eight total) in which the ambiguous stimulus was chosen (in the ambiguous-novel pairings) by Migwan across probe trials by season.

**Figure 7 behavsci-07-00051-f007:**
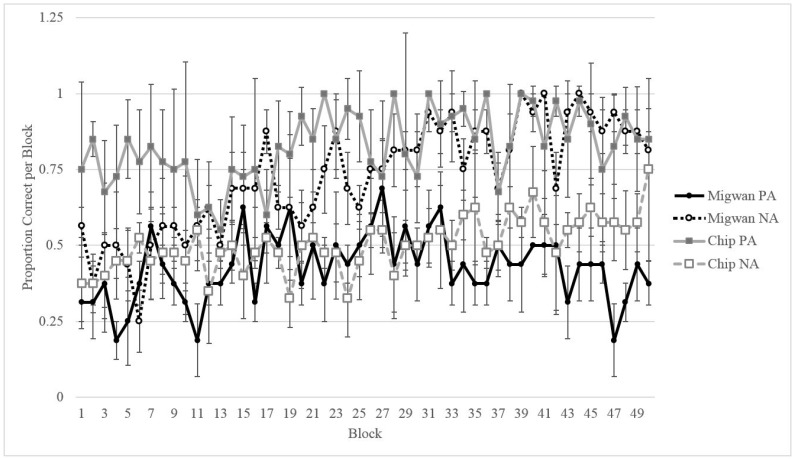
Comparison of Chip and Migwan’s performance on each trial type across their first 200 sessions of training. Data is shown in blocks of four sessions.

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
