# Peer review of "Ambiguous Results When Using the Ambiguous-Cue Paradigm to Assess Learning and Cognitive Bias in Gorillas and a Black Bear"

_behavsci, 2017, doi:10.3390/bs7030051_

Round 1

Reviewer 1 Report

General

With many of these ambiguous-testing paradigms, the lack of a reward seems to be insufficient to put the animal in a “negative state”, and therefore the authors should address this issue in their discussion. For example, the limitations of a lack of reward for S- and the potential results that could be found if the S- was a punishment (shock, air puff, etc.). The interpretation of the results should have a higher level of caution, as the goal was to examine seasonal influence on affective states in captive gorillas, but the animals were only testing in winter. Therefore, the objective should be adjusted, as the authors are not demonstrating a study which addresses seasonal changes or environmental complexity/size impacting one species’ performance on this task. The gorillas were testing in only one season and the bear never changed environments (therefore it is inappropriate to compare to the gorillas). In addition, there is a lack of data to support that performance in this ambiguous-test relates to the affective state of these animals (i.e., other metrics of animal welfare were not collected). This study essentially examined (in two non-related studies) differences in learning mechanisms. They provided results for the difference in learning NA versus PA trials, and the approach-approach vs avoid-avoid issue (as well as preference for familiar cues) in one single animal per species. No evidence was given to make the argument of seasonal or environmental contexts influencing animals’ affective state.

Specific

Line 30-33: Could the authors provide examples of PA and NA (such as near positive location paradigm).

Line 64-74: While the placement of this information belongs in the methods, it is understandable why it is in the introduction. (Unless this is a journal style preference, there is a lot of information on the methods in this introduction that should be removed). Could the authors condense these paragraphs into one brief overview, such as:

There is evidence that providing animals with larger, more enriched enclosures may elicit positive cognitive biases (in rats [16]). Captive European starlings exhibited more optimistic interpretations of ambiguous stimuli after they had been housed in larger cages that also contained more enrichment items (such as water baths, perches, and bark chips) compared to when they had been housed in smaller cages lacking any additional enrichment items ─ although, in this case, space and enrichment were confounded [17]. The cyclical changes in environment that captive animals encounter may be due to the seasonal changes in climate, seasonal changes in visitor numbers, or seasonal changes in husbandry routines. In Experiment 1 of the present study, three male gorillas were moved between indoor and outdoor enclosures, with varying degrees of enrichment complexity, based on the season. It may be that these seasonal changes in the gorilla habitat cause similar patterns of cognitive bias as those described by Matheson et al. [17], with the gorillas displaying more positive biases when in the outdoor habitat than when in the indoor habitat.

In Experiment 2 of the present study, the American black bear focal animal did not experience changes to habitat access, but rather changes to visitor density, with warmer weather during the summer drawing larger crowds than cooler temperatures during the fall and winter. To date, there has been only one study to investigate cognitive biases in bears….”

Lines like “The outdoor habitat is larger and contains natural foraging opportunities in the form of plants (living grasses, bushes, and trees), in addition to many of the typical enrichment items that they may be offered in the indoor habitat (e.g., cardboard, toys, cut foraging material)” are too much detail on the methods to be placed in the introduction.

Lines 117-127: The authors need to have a stronger concluding sentence. Given the argument of bears’ cognitive capacity due to foraging, do the authors believe that the bears will outperform the gorillas?  What are the arguments that gorillas could outperform bears (i.e., why is it assumed that gorillas can handle this task? What is their foraging strategy?)

Figure 1. Can this image have labeled columns for “N”, “P”, “NA, NP” and “novel”? Were the first three images were paired with N and P, or were 2 of the first 3 labeled as N and P themselves? Perhaps examples of what the gorilla’s saw at each trial could help the reader to understand?

Lines 144-152/158-162: The training is a bit confusing and could be spelled out more clearly (perhaps by adjusting Figure 1). The text reads as if the gorillas were given a choice of N vs P, N vs. NA, P vs PA, and PA vs NA. I believe that the gorillas were trained under N v P, P v A, and N vs A; the “TEST” was introducing A v novel, correct? If so, this should be much easier to understand.

 What was the inter-trial-interval (how long before advancement to next trial) and was ITI difference for correct versus incorrect choices?

Would the gorillas be rewarded for choosing either the novel probe or the A cue in the test?

Line 231-232: “Perhaps this made it easier to monitor the whereabouts of other group members.” What do the authors mean by this? Is there evidence that monitoring social group can influence the affective state of a gorillas?

Line 253: Can the authors find alternative evidence for why training 2_D images to apes is difficult (instead of an “unpublished data” reference)?

Line 284-286: How did Migwan make a choice? Nose press to touchscreen?

Figure 6. Remind the reader that Migwan was choosing A versus novel probe for these tests in the figure caption.

Lines 447-454: This description of how animals must work through stimuli whicha re never or sometimes rewarded in NA pairings should go in the introduction or the methods as it sets up the objective of this test.

Author Response

General

With many of these ambiguous-testing paradigms, the lack of a reward seems to be insufficient to put the animal in a “negative state”, and therefore the authors should address this issue in their discussion. For example, the limitations of a lack of reward for S- and the potential results that could be found if the S- was a punishment (shock, air puff, etc.). The interpretation of the results should have a higher level of caution, as the goal was to examine seasonal influence on affective states in captive gorillas, but the animals were only testing in winter. Therefore, the objective should be adjusted, as the authors are not demonstrating a study which addresses seasonal changes or environmental complexity/size impacting one species’ performance on this task. The gorillas were testing in only one season and the bear never changed environments (therefore it is inappropriate to compare to the gorillas).

We have focused the MS instead on the potential for the training trials of the ambiguous cue paradigm to indicate both learning strategies and stable cognitive biases of the subjects. In this regard, we think it is interesting to compare species regardless of similarities or differences in environmental conditions. Bears have previously participated in tasks similar to those presented to great apes and it is of interest to compare their learning, especially given that both are large-brained mammals but primates are highly social and bears are relatively asocial. Bears may also experience similar levels of foraging complexity. Thus, testing bears and great apes can allow us to examine the role of factors such as sociality and foraging complexity in predicting cognitive skills. Our MS does not depend upon the effects of seasonal changes. Indications of consistency in cognitive bias across test days are of interest at an individual level. We hope these points come through more clearly in the revision.

 In addition, there is a lack of data to support that performance in this ambiguous-test relates to the affective state of these animals (i.e., other metrics of animal welfare were not collected). This study essentially examined (in two non-related studies) differences in learning mechanisms. They provided results for the difference in learning NA versus PA trials, and the approach-approach vs avoid-avoid issue (as well as preference for familiar cues) in one single animal per species. No evidence was given to make the argument of seasonal or environmental contexts influencing animals’ affective state.

The reviewer is correct that there are limitations on what we can conclude from the existing data. We feel we have been appropriately cautious in our conclusions. However, we do think that this paradigm, not previously used to assess cognitive bias, can be a useful addition to the literature on affective states in captive animals, so we think it makes a contribution insofar as it inspires other researchers to apply this method to test other subjects. We also think the individual differences in learning are of interest because they suggest that learning biases are not species-specific.

Specific

Line 30-33: Could the authors provide examples of PA and NA (such as near positive location paradigm).

The stimuli appear in Figure 1. A PA or NA trial would involve the pairing of two of these stimuli (Different for each subject). We are not clear on what the reviewer means by near positive location paradigm.

Line 64-74: While the placement of this information belongs in the methods, it is understandable why it is in the introduction. (Unless this is a journal style preference, there is a lot of information on the methods in this introduction that should be removed). Could the authors condense these paragraphs into one brief overview, such as:            

There is evidence that providing animals with larger, more enriched enclosures may elicit positive cognitive biases (in rats [16]). Captive European starlings exhibited more optimistic interpretations of ambiguous stimuli after they had been housed in larger cages that also contained more enrichment items (such as water baths, perches, and bark chips) compared to when they had been housed in smaller cages lacking any additional enrichment items ─ although, in this case, space and enrichment were confounded [17]. The cyclical changes in environment that captive animals encounter may be due to the seasonal changes in climate, seasonal changes in visitor numbers, or seasonal changes in husbandry routines. In Experiment 1 of the present study, three male gorillas were moved between indoor and outdoor enclosures, with varying degrees of enrichment complexity, based on the season. It may be that these seasonal changes in the gorilla habitat cause similar patterns of cognitive bias as those described by Matheson et al. [17], with the gorillas displaying more positive biases when in the outdoor habitat than when in the indoor habitat.

In Experiment 2 of the present study, the American black bear focal animal did not experience changes to habitat access, but rather changes to visitor density, with warmer weather during the summer drawing larger crowds than cooler temperatures during the fall and winter. To date, there has been only one study to investigate cognitive biases in bears….”

Lines like “The outdoor habitat is larger and contains natural foraging opportunities in the form of plants (living grasses, bushes, and trees), in addition to many of the typical enrichment items that they may be offered in the indoor habitat (e.g., cardboard, toys, cut foraging material)” are too much detail on the methods to be placed in the introduction.

We thank the reviewer for these suggestions. Much of this information was moved to the methods section and the section condensed as suggested.

Lines 117-127: The authors need to have a stronger concluding sentence. Given the argument of bears’ cognitive capacity due to foraging, do the authors believe that the bears will outperform the gorillas?  What are the arguments that gorillas could outperform bears (i.e., why is it assumed that gorillas can handle this task? What is their foraging strategy?)

We have revised this section to read: The capacity of bears to outperform apes in cognitive tests supports recent conjecture that foraging complexity is potentially more important than sociality in driving the evolution of certain aspects of complex cognition [27,28] given that bears experience low levels of sociality but varying levels of foraging complexity whereas apes experience high levels of sociality and foraging complexity. Both species can be described are generalists that exploit a patchily distributed diet and engage in extractive foraging [29], qualifying them as experiencing complex foraging demands. We were interested in cognitive bias at the individual level and all subjects were of interest in this regard given their unique housing situations (a bachelor group of gorillas and a solitary black bear).

Figure 1. Can this image have labeled columns for “N”, “P”, “NA, NP” and “novel”? Were the first three images were paired with N and P, or were 2 of the first 3 labeled as N and P themselves? Perhaps examples of what the gorilla’s saw at each trial could help the reader to understand?

The items were assigned differently to N, P and A categories depending upon the subject, as explained in the Procedure section. So each subject saw a different combination of stimuli for each type of trial. But all that they saw were two of these stimuli presented side by side on the screen for each trial. We didn’t think that an additional figure added significant clarity beyond what was presented in Figure 1.

Lines 144-152/158-162: The training is a bit confusing and could be spelled out more clearly (perhaps by adjusting Figure 1). The text reads as if the gorillas were given a choice of N vs P, N vs. NA, P vs PA, and PA vs NA. I believe that the gorillas were trained under N v P, P v A, and N vs A; the “TEST” was introducing A v novel, correct? If so, this should be much easier to understand.

We apologize for the confusion. The reviewer is correct about how the trials were presented, consistent with other studies using the ACP. We hope the revision is clearer.

 What was the inter-trial-interval (how long before advancement to next trial) and was ITI difference for correct versus incorrect choices?

The ITI was 750 MS and did not vary between correct and incorrect choices. We have added this information to the MS.

Would the gorillas be rewarded for choosing either the novel probe or the A cue in the test?

Thank you for prompting us to clarify this. We included a sentence clarifying that the gorillas were rewarded for either choice.

Line 231-232: “Perhaps this made it easier to monitor the whereabouts of other group members.” What do the authors mean by this? Is there evidence that monitoring social group can influence the affective state of a gorillas?

The thought here was that when socially compressed, it may have actually reduced stress as they are better able to keep track of each other’s movements. If they know where their rivals are located, they would be better prepared for a potentially antagonistic encounter. This sentence has been clarified.

Line 253: Can the authors find alternative evidence for why training 2_D images to apes is difficult (instead of an “unpublished data” reference)?

We were referring to the specific gorillas that we tested so we refer only to our own work here. We do not wish to generalize to other gorillas, as we do not think that such a generalization would be accurate or appropriate.

Line 284-286: How did Migwan make a choice? Nose press to touchscreen?

The reviewer is correct that Migwan used her nose to make selections on the touchscreen computer. This information has been included.

Figure 6. Remind the reader that Migwan was choosing A versus novel probe for these tests in the figure caption.

We thank the reviewer for this suggestion and have included this information in the caption.

Lines 447-454: This description of how animals must work through stimuli which a re never or sometimes rewarded in NA pairings should go in the introduction or the methods as it sets up the objective of this test.

This issue is addressed in lines 39-49 in the introduction.

We hope that we have successfully resolved the points raised by the reviewer.

Reviewer 2 Report

This is an interesting study. I do not have any comments or suggestions for the authors.

Author Response

Reviewer 2 did not provide any comments for us to respond to.

Round 2

Reviewer 1 Report

The authors had made all relevant changes suggested to the manuscript.